# Vitamin D Repletion and AA/EPA Intake in Children with Type 1 Diabetes: Influences on Metabolic Status

**DOI:** 10.3390/nu14214603

**Published:** 2022-11-01

**Authors:** Silvia Savastio, Erica Pozzi, Valentina Mancioppi, Valentina Boggio Sola, Deborah Carrera, Valentina Antoniotti, Paola Antonia Corsetto, Gigliola Montorfano, Angela Maria Rizzo, Marco Bagnati, Ivana Rabbone, Flavia Prodam

**Affiliations:** 1SCDU of Pediatrics, Department of Health Sciences, University of Piemonte Orientale, 28100 Novara, Italy; 2Department Dietetic and Clinical Nutrition, University Hospital of Novara, 28100 Novara, Italy; 3Laboratory of Membrane Biochemistry and Applied Nutrition, Department of Pharmacological and Biomolecular Sciences, Università degli Studi di Milano, 20133 Milan, Italy; 4Clinical Biochemistry, University Hospital of Novara, 28100 Novara, Italy; 5Department of Health Sciences, University of Piemonte Orientale, 28100 Novara, Italy; 6Endocrinology, Department of Translational Medicine, University of Piemonte Orientale, 28100 Novara, Italy

**Keywords:** omega-3, EPA, DHA, omega-6, arachidonic acid, type 1 diabetes, *C*-peptide, vitamin D, AA/EPA ratio

## Abstract

Our study aimed to show a relationship between metabolic control, vitamin D status (25OHD), and arachidonic acid (AA)/eicosapentaenoic acid (EPA) ratio in children with type 1 diabetes (T1D). The secondary aim was to evaluate dietary intake and the presence of ketoacidosis (DKA) at the onset of T1D. Methods: A cohort of 40 children with T1D was recruited, mean age 9.7 years (7.1; 13), with onset of T1D in the last 5 years: some at onset (*n*: 20, group A) and others after 18.0 ± 5 months (*n*: 20; group B). Twenty healthy children were compared as control subjects (CS). Dietary intakes were assessed through a diary food frequency questionnaire. Moreover, dried blood spots were used to test AA/EPA ratio by gas chromatography. Results: T1D children had a lower percentage of sugar intake (*p* < 0.02) than CS. Furthermore, group B introduced a greater amount of AA with the diet (g/day; *p* < 0.05) than CS (*p* < 0.01) and group A (*p* < 0.01). Children with an AA/EPA ratio ≤ 22.5 (1st quartile) required a lower insulin demand and had higher 25OHD levels than those who were in the higher quartiles (*p* < 0.05). Subjects with DKA (9/40) had levels of 25OHD (*p* < 0.05) and *C*-peptide (*p* < 0.05) lower than those without DKA. Moreover, analyzing the food questionnaire in group A, subjects with DKA showed a lower intake of proteins, sugars, fiber (g/day; *p*< 0.05), vitamin D, EPA, and DHA (g/day; *p* < 0.01) compared to subjects without DKA. Non-linear associations between vitamin D intake (*p* < 0.0001; r2:0.580) and linear between EPA intake and *C*-peptide (*p* < 0.05; r: 0.375) were found in all subjects. Conclusions: The study shows a relationship between vitamin D status, AA/EPA ratio, and metabolic state, probably due to their inflammatory and immune mechanisms. A different bromatological composition of the diet could impact the severity of the onset.

## 1. Introduction

In managing type 1 diabetes (T1D), there is a general agreement on the benefits of the Mediterranean diet in improving glycemic control [1,2]. Nutritional deficiency or insufficiency could impair the immune system. In particular, vitamin D (assayed as serum 25OHD) and omega-3 long-chain polyunsaturated fatty acids (Ω-3) appear to maintain or increase immune functions and promote anti-inflammatory actions with the inhibition of pro-inflammatory mediators [3]. Moreover, some authors showed that the administration of Ω-3 and vitamin D together might improve the anti-inflammatory therapy in islet allotransplantations [4].

Low vitamin D values appear to be associated with poor glycemic control in T1D [5,6,7]. Furthermore, the prevalence of vitamin D deficiency is higher in T1D children than in healthy controls [8], and birth cohort studies have analyzed the association between 25OHD values and T1D risk with contrasting results [9,10,11]. However, one Mendelian randomization study showed an association between vitamin D deficiency and T1D risk [12].

Several studies showed an immunological effect of vitamin D in T1D subjects. In detail, the supplementation with 2000 U/die of cholecalciferol for 12 months determined a significant increase in regulatory T lymphocytes cells (Tregs) percentages in new-onset T1D [13]. Similarly, T lymphocyte profiles changed in male adults with T1D after 4000 IU/die of cholecalciferol for three months [5]; moreover, Treg suppressive capacity increased in T1D subjects after 70 IU/Kg body weight/day of cholecalciferol was detected [14]. Furthermore, also our group demonstrated a link between serum 25OHD levels, lymphocyte T helper 17 (Th17), and Treg with a costimulatory receptor (ICOS+) expressed on activated effector T cells in subjects with T1D and their siblings. Poor vitamin D levels in siblings cause a dysregulated immune response that can be partially improved by cholecalciferol therapy, suggesting a vitamin D role in phases before the beta cell destruction [15]. According to these and other findings, very recently, the Consensus Statements from the second and third International Conference on Controversies in Vitamin D underlined that increasing evidence suggests that vitamin D in autoimmune conditions, including T1D, could have a preventive rather than therapeutic role [16,17].

Eicosapentaenoic acid (EPA) and docosahexaenoic acid (DHA) are two master regulators of immune functions having T-cell inhibitory actions [18,19]. Ω-3 exerts potent anti-inflammatory properties generating distinct PUFA-derived metabolites, such as resolvins, maresins, and protectins, named specialized pro-resolving mediators. They act as strong anti-inflammatory agents inhibiting the production of inflammatory eicosanoids leukotrienes, pro-inflammatory cytokines, chemokines, platelet activating factors, and adhesion molecules. Moreover, Ω-3 increases anti-inflammatory cytokine production, such as interleukin 10 [2,20].

In vivo studies in NOD mice showed that a long-term dietary intervention with Ω-3 reduced T1D severity and incidence, decreasing pro-inflammatory T-cell subsets (Th1, Th17) and cytokines and reciprocally increasing anti-inflammatory T-cell subsets (Th2, Treg) [21].

In humans, protective effects of Ω-3 have been reported in chronic and autoimmune disorders such as T1D. A long-term Ω-3 dietary intake at 1 year of age was linked with a reduction of the risk to developing T1D in general population as well as islet autoimmunity in children at risk for T1D [22,23]. In addition, some clinical studies in T1D showed favorable outcomes on the persistence of *C*-peptide and lower glycated hemoglobin with a sustained intake of Ω-3 and/or vitamin D [24,25].

In contrast, arachidonic acid (AA, Ω-6) was described as having an opposite effect on Treg, Th17 cells, and Th1/Th2 ratio, functioning as the precursor of pro-inflammatory eicosanoids and therefore being toxic to the beta cells [26]. Subjects with T1D and other autoimmune conditions frequently presented a very high AA/EPA ratio, reflecting a diet-related pro-inflammatory baseline condition due to an imbalance of these essential lipids. This condition could predispose or trigger the subsequent development of autoimmunity [27].

Since the recent results of the VITAL study on the protective role on incident autoimmune diseases of the co-administration of cholecalciferol and Ω-3 (EPA and DHA) for 5 years in middle-aged adults [28], several hypotheses have been discussed regarding the mechanistic interplay of these two players in the immunity risk [29]. On the other hand, Ω-3 are essential nutrients, but their dietary contribution in relation to dietary intake and/or supplementation with cholecalciferol and phenotypic characteristics of autoimmune diseases such as T1D has not been established so far.

Indeed, our study aimed to investigate the dietary habits in T1D children and the relationship between vitamin D and Ω-3 dietary intake, 25OHD levels, AA/EPA ratio, and metabolic parameters (glycated hemoglobin HbA1c, insulin demand) in children with T1D. We also aimed to evaluate dietary intake in a global view and the severity of the disease at the onset of T1D.

## 2. Materials and Methods

### 2.1. Patients and Methods

Forty subjects (male 19/female 21) with T1D onset in the last five years were enrolled at the Division of Pediatrics of our hospital in 2019 in the proof-of-concept part of a trial on vitamin D and Ω -3 supplementations in the Mediterranean diet during the 1st year of overt type 1 diabetes (NCT: NCT03911843) [30].To identify if differences are related to the time from the onset, a 1:1 approach was used, and 20 subjects at the time of onset (group A) within a maximum of two months and 20 after 18 ± 5 months from onset (group B) were recruited. In addition, a further group of 20 healthy children who were referred to Pediatric Surgery or Orthopedic Clinics for minor surgery or mild trauma and matched for sex and age were recruited as controls (CS) for the blood Ω-3 measurements. The study was approved by the Ethical Committee of Novara (protocol number 143/17) and conformed to the guidelines of the European Convention of Human Rights and Biomedicine for Research in Children. All patients and controls were included in the study after written consent from the parents.

Patients and controls treated with drugs that could affect immunity or glucose metabolism, including corticosteroids, ciclosporin, and tacrolimus, were excluded. Subjects with celiac disease or thyroiditis were not excluded. The diagnosis of T1D was performed according to the American Diabetes Association criteria [31]. Micro or macrovascular complications were defined according to the ISPAD criteria [32]. Children’s height, weight, and BMI were evaluated using the Italian growth charts (Cacciari 2006).

At the onset of the disease, clinical data, presence of diabetic ketoacidosis, HbA1c%, insulin requirement, 25OHD levels, thyroid function, antibody title of GADA, IAA, IA-2, antibodies for celiac disease, and lipid profile were collected. Evidence of ketoacidosis was assessed according to the ISPAD criteria [31], and severe DKA was considered if pH < 7.1. The vitamin D status (25OHD plasma levels) was classified according to the Endocrine Society criteria, graded as sufficiency for ≥30 ng/mL (≥75 nmol/L), insufficiency 20–29 ng/mL (50–74.9 nmol/L), and deficiency < 20 ng/mL (<50 nmol/L) [33].

All T1D subjects with vitamin D insufficiency/deficiency were supplemented with cholecalciferol 1000 IU/die or 2000 IU/die for 8–12 weeks as indicated by guidelines [34,35,36].

At enrollment, metabolic parameters (HbA1c%, *C*-peptide, insulin requirement) were evaluated. Fatty acids percentages (%) (AA, EPA, DHA percentages, and respective AA/EPA ratio) on dry blood spots and vitamin D serum values were also determined.

### 2.2. Dietary Assessment

At enrollment, a dietician collected nutritional history and performed an educational nutritional training according to the Mediterranean diet [1]. Dietary intakes were assessed based on a food diary from 5 different days. The amounts of the various nutrients were calculated with the support of the Food Composition Database for Epidemiological Studies (BDA version 1-2015) based on INRAN Tables (http://www.crea.gov.it; accessed date 1 January 2020). Protein (g/day), energy (Kcal/day), cholesterol (mg/day), total and simple carbohydrates (g/day), dietary fiber (g/day), vitamin D (g/day), polyunsaturated fatty acids (g/day), arachidonic acid (AA g/day), EPA (g/day), and DHA (g/day) were extracted. Moreover, the percentage of nutrients in relation to the total caloric share was evaluated.

### 2.3. Assays

Plasma glucose levels (mg/dL; 1 mg/dL:0.05551 mmol/L) were measured by the gluco-oxidase colorimetric method (GLUCOFIX, by Menarini Diagnostics, Florence, Italy). HbA1c% levels were measured through the high-performance liquid chromatography (HPLC), using a Variant machine (Biorad, Hercules, CA, USA); intra- and inter-assay coefficients of variation are, respectively, lower than 0.6% and 1.6%. Linearity is excellent from 3.2% (11 mmol/mol) to 18.3% (177 mmol/mol). The presence and titration of antibodies GAD65, IA2, and IAA, expressed in IU/mL, was carried out by IRMA (Immunoradiometric Assay) with analytical coefficient of analysis of 13%, 8.4%, and 13%, respectively. The levels of circulating *C*-peptide, expressed in ng/mL, were measured on citrate or heparinized plasma, both by chemiluminescent “sandwich” immunoassay (DiaSorin Liaison, Novara, Italy) and by immunochemiluminescence (CLIA) with automatic analyzer Immulite 2000, Medical Systems, with a coefficient of variability of 7.40%.

The semi-quantitative determination of TGA, AGAD-G, and AGAD-A, expressed in IU/mL, was carried out on serum by QUANTA Flash, a rapid response iCLIA performed on the BIO-FLASH instrument; the coefficient of analytical variability is, respectively, 5.5%, 6.7%, and 4.3%. The quantitative determination of the anti-TG and anti-TPO autoantibodies, expressed in IU/mL, of the TSH, which is expressed in µUI/mL, and of the fT4, expressed in ng/dL, was performed by a competitive immunoassay using immuno-chemiluminescence technique direct using the Siemens ADVIA centaur XPT system.

25OHD serum levels (nmol/L) were measured with a direct competitive chemiluminescent immunoassay (Liaison Test 25OHD total, DiaSorin Inc, Stillwater MNUSA). CV for inter-assay analyses was 10%. The method is conformed to International DEQAS standards.

Whole-blood fatty acid composition (AA, EPA, and DHA as percentages of total fatty acids and respective AA/EPA ratio) was determined by gas chromatography using dried blood spots testing [37].

### 2.4. Statistical Analysis

The data were non-normally distributed according to the Shapiro–Wilk test, so a non-parametric approach was used. Data are expressed as median (IQR) or percentage (%), as appropriate. The differences between the two groups were evaluated for the continuous variables through the Mann–Whitney U-test; while the differences between tree groups (in particular, Groups A, B, and CS) were performed with the Kruskal–Wallis test and post hoc analysis. Chi-square test was used for comparison of nominal variables between groups. The best association between variables was evaluated by studying curves. The association between the variables was evaluated according to Pearson test after logarithmic transformation of the parameters when necessary. The AA/EPA ratio was stratified in quartiles, and the first quartiles were considered as the threshold for metabolic parameter assessments. Trend evaluation across 25OHD levels was performed at onset through multinomial regression analyses.

Significant *p*-values were less than 0.05. All statistical analyses were performed using SPSS 22.0 (SPSS Inc., Chicago, IL, USA).

## 3. Results

The clinical data of the 40 enrolled patients are reported in Table 1.

Among them, five subjects (10%) had celiac disease and were on a gluten-free diet, and one (2.5%) congenital hypothyroidism under replacement therapy.

All 40 T1D subjects, mean age 9.7 years (7.1; 13), completed the food questionnaire. CS were matched for age and sex with T1D. Group A had age, weight, and insulin therapy dosages (IU/Kg/day) lower (*p* < 0.05) than Group B. Regarding nutrient intake at baseline, Group B showed a lower intake of sugars than CS (%, *p* < 0.01) and higher AA introduction (g/day *p* < 0.05) than CS. Group A had AA intake lower than B without other differences between them (*p* < 0.01) (Table 1).

Moreover, evaluating biochemical parameters, Group A showed lower 25OHD values when compared with Group B (*p* < 0.05) without any other differences (Table 1).

No differences were detected in results excluding patients with celiac disease and congenital hypothyroidism. The differences in 25OHD levels were detected despite similar dietary intake of cholecalciferol/ergocalciferol.

AA/EPA ratio and EPA and DHA percentages on dry spot were similar between subjects with T1D and CS.

The overall assessment of all T1D patients demonstrated that AA/EPA value ≤ 22.5 (1st quartile) was associated with lower insulin requirements (*p* < 0.05) and higher 25OHD levels (*p* < 0.05) when compared with AA/EPA ratio > 22.5 (Figure 1). Similar data were shown analyzing Group A individually but not Group B.

Regarding 25OHD levels, at baseline, patients with T1D showed insufficiency and deficiency in 29.6% and 51.8% of cases, respectively. No difference in HbA1c and insulin therapy dosage (IU/Kg/day) across 25OHD levels was observed.

At baseline, no significant correlations were found between vitamin D, AA/EPA ratio, and metabolic parameters (*C*-peptide, HbA1c or IU/Kg/day) in both groups. Non-linear associations between vitamin D intake and EPA intake (*p* < 0.0001; r2:0.580, cubic regression) and positive associations between EPA intake and *C*-peptide (*p* < 0.05; r: 0.375) were found in all subjects together and in Group B with similar significant values.

### Dietary Intake and DKA at T1D Onset

At the onset of T1D, 9/40 T1D children presented DKA and 4/40 severe DKA (PH < 7.1 mmol/L). Subjects with DKA showed age (*p* < 0.05), 25 OH vitamin D (*p* < 0.05), and *C*-peptide (*p* < 0.05) values lower than patients without this adverse event (Table 2).

Subjects with severe DKA showed higher glucose values (*p* < 0.01) and lower 25OHD (*p* < 0.0001) when compared with patients without this condition. Patients with vitamin D levels (25OHD) ≤ 10 ng/mL showed more severe ketoacidosis with lower pH and *C*-peptides values (*p* < 0.05) compared to subjects with a better vitamin D status (Figure 2).

There was a widespread vitamin D insufficiency in T1D children (81.5%) at the onset. Severe deficiency ≤ 10 ng/mL (25 nmol/L) was present in 14.8% of subjects. The lowest levels were present in winter and spring 16.7 (11.2; 23.1) ng/mL vs. summer and autumn 27.1 (19.3; 33.1) ng/mL, as expected (χ2 6.724; *p* < 0.01).

Moreover, by analyzing dietary intakes in relation to DKA status in patients of Group A, differences were found. Subjects with DKA showed a lower intake of protein (g/day), sugars (g/day and %), fiber (g/day) (*p* < 0.05), and vitamin D (µg/day; *p* < 0.01). Interestingly, subjects with DKA showed a lower intake of EPA and DHA (*p* < 0.01) than non-DKA patients. Finally, a higher AA/EPA ratio, even if not significant, was found (Table 3).

At the onset, relationships between 25OHD and pH (*p* < 0.001; r: 0.498) and 25OHD and whole blood AA/EPA ratio (*p* < 0.01; r: −0.599) were observed. Furthermore, vitamin D intake was negatively correlated with insulin therapy dosage (IU/Kg/day) at hospital discharge.

## 4. Discussion

Lifestyle with diet and physical activity are cornerstones of the treatment of T1D beyond insulin therapy. Medical nutrition therapy and nutritional education are recommended in pediatric age to optimize glucose outcomes and provide appropriate energy intake and nutrients for optimal growth and development. However, deciphering in T1D the role of tailored nutrition one step further in carbohydrate counting is still a challenge, and we addressed exploring some dietary aspects in the first phases of the disease.

First, we showed that patients 18 months after the onset of the disease paid more attention to diet, reporting a lower sugar and higher AA intake than CS. A little higher intake of EPA, DHA, protein, and fiber were also present vs. CS (not significant). The minimized consumption of foods containing simple sugars is in line with international guidelines [31]. Instead, a diet rich in fibers and highly available proteins, particularly from vegetable sources, such as legumes, should be encouraged to improve satiety and reduce cardiovascular risks [31]. Although we could not detect the protein source, our patients followed all these suggestions. These results were likely reached due to dietician support and counseling that carried out an efficacious nutrition education for children and their parents [31].

Diversely, evaluating dietary assessment in patients at the onset of the disease, we showed no differences with CS. Moreover, subjects with DKA had a lower intake of protein, sugars, fibers, vitamin D, EPA, and DHA than non-DKA patients. All these findings could result from ketoacidosis, a metabolic state characterized by nausea, anorexia, constipation, headaches, and fatigue, all adverse events affecting hunger and prospective food consumption [38,39].

On the other hand, an unbalanced diet prior to the onset of the disease could be a risk factor not only for an increased incidence, as reported by several epidemiological studies [40], but also for the severity of the first clinical manifestation if we consider as generally malnutrition impacts on all the diseases’ outcomes [41].

Among nutritional deficiencies, that of vitamin D is one under the research magnifying glass as a trigger in T1D development [42,43].

Although the exact mechanism is still unclear, vitamin D metabolites have immune-modulating and anti-inflammatory actions, inhibiting T-cell activation and the modifying T-cell cytokine profile from a proinflammatory to an anti-inflammatory one [15,42]

Several studies investigated the link between 25OHD values and T1D risk [44]. In 2018, the TEDDY study [9] found in 8676 children at increased genetic risk for T1D an association between low vitamin D values during childhood and an increased risk of islet autoimmunity. If observational findings give a strong suggestion about the involvement of vitamin D system in the autoimmunity process, benefits of vitamin D supplementation in general or in a specific population to prevent or delay T1D onset are still conflicting due to several biases, heterogeneous regimens, and windows selected for the interventions [10,11]. However, vitamin D supplementation with alfacalcidol or cholecalciferol 2000 IU/die for about 6 to 18 months appears to attenuate the natural history of the disease with significant positive effects on *C*-peptide levels, as observed in a recent systematic review [45]. In agreement with these data, we found an association between vitamin D intake, *C*-peptide levels, and insulin requirement at hospital discharge in patients with T1D. Our results suggest a role of vitamin D in the preservation of beta cell mass likely through immune-modulating and anti-inflammatory actions [44].

The high prevalence of hypovitaminosis D at the onset shown in our subjects is widely reported in the pediatric population with T1D [46,47]. We confirmed again these results at the onset despite similar dietary intake of vitamin D with respect to later on in the study. These results suggest that natural food sources of vitamin D are limited, and other more impacting factors influence vitamin D repletion. Although we do not have precise data on the UVB irradiation of our cohort, they lived at the same latitude and were normal-weight, supporting the hypothesis that players such as inflammation and immunity factors could modulate circulating 25OHD levels at the onset of T1D [46,47]. On the other hand, vitamin D supplementation determined an increase in 25OHD levels in group B as expected, while vitamin D food fortification may be a crucial strategy to increase dietary vitamin D intakes and improve vitamin D status mainly in at-risk populations [33].

Moreover, vitamin D values appeared to be lower in T1D with diabetic ketoacidosis at the onset, suggesting a reduced immunomodulating action of vitamin D in this metabolic condition [48]. The expression of the vitamin D receptor on pancreatic beta cells appears to have a role in insulin secretion, and low vitamin D levels could facilitate the onset of diabetic ketoacidosis [48].

However, on the contrary, low 25OHD levels could result from DKA since inactivation of the 1-alpha-hydroxylase enzyme and increased renal excretion of the vitamin D binding protein is described in this condition [48].

Recently, some studies have outlined the importance of other nutrients such as omega-3, among others, in preserving *C*-peptide and improving glycated hemoglobin [24,25]. In our cohort, the intake of EPA or EPA plus DHA has been positively associated with *C*-peptide at follow-up in children with T1D, supporting their role in the β-cell preservation and function.

Although we found AA/EPA ratio and EPA and DHA percentages in whole blood to be similar between subjects with T1D and CS, patients with AA/EPA value ≤ 22.5 (1st quartile) showed higher 25OHD levels and lower insulin demand at discharge than others with a higher ratio. Since omega-6 and omega-3 balance modulates cytokine profiles [22,23,28], they could affect inflammatory processes associated with the early phases of autoimmunity and its progression. However, despite the supposed protective role of vitamin D and omega-3, clinical trials are still inconclusive about a clear benefit on children with or at risk for T1D [26].

This study had several limitations. First, we enrolled a limited number of subjects with T1D, both at the onset and with overt disease. Second, we used only a food diary because a validated frequency questionnaire for children with T1D is not available. Third, our study cannot establish causal relationships or strong associations between clinical outcomes and vitamin D or fatty acids because of its design. Finally, we did not evaluate PTH, which is usually increased in autoimmune diseases. However, these data are preliminary to designing further studies investigating the role of diet and nutrients in T1D and strengthening new, intriguing data in adults. As cited above, the VITAL study recently showed how vitamin D and omega-3 supplementations for five years reduced the risk of autoimmune diseases in adults. However, these data are not conclusive about an immunity protection in younger people when genetic and environmental risks could be different [49]. Further-powered and sufficiently prolonged RCTs on vitamin D and omega-3 supplementation in the pediatric age are mandatory to deciphering if these player are significantly able to decrease the risk of T1D in children at risk or contribute to the improvement of the metabolic control in those affected.

## 5. Conclusions

In conclusion, our study confirmed a significant presence of hypovitaminosis D in children with T1D. AA/EPA ratio and vitamin D levels in a normal range seem to be associated with better glucose homeostasis and clinical outcomes.

These preliminary findings support the hypothesis of a role of vitamin D and omega-3 and -6 polyunsaturated fatty acids in the crosstalk of the immune and inflammatory responses. Tailored medical nutrition therapy inspired by immuno-nutrition is a challenge for the work-up of T1D and needs further research.

## Figures and Tables

**Figure 1 nutrients-14-04603-f001:**
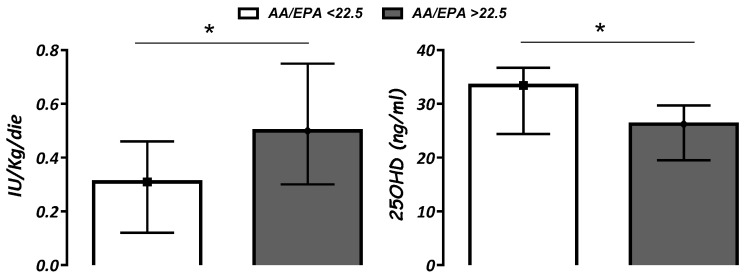
Insulin requirement (IU/Kg/day) and serum vitamin D (25OHD; ng/mL) levels according to AA/EPA ratio below or above 22.5 (1st quartile). Data are expressed as median (IQR). * *p* < 0.05.

**Figure 2 nutrients-14-04603-f002:**
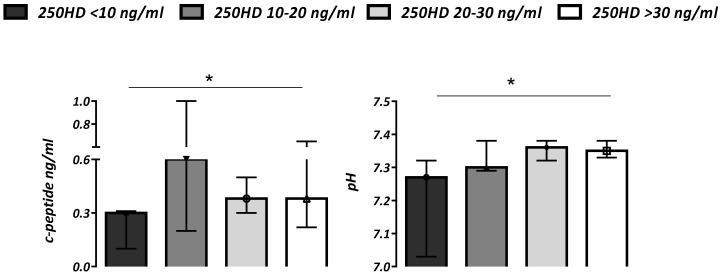
*C*-peptide (ng/mL) and pH values according to vitamin D status (serum 25OHD ng/mL) in all subjects. Data are expressed as median (IQR). * *p* for trend < 0.05.

**Table 1 nutrients-14-04603-t001:** Auxologic and metabolic parameters and dietary assessment in type 1 diabetes patients (T1D) and control subjects (CS) at the enrolment (T0).

	T1D Group A	T1D Group B	*p* *	CS	*p* °
Number	20	20		20	
Age (y)	8.1 (5.7; 10.6)	10.5 (7.9; 14)	<0.05	11.5 (9.3; 13.6)	0.155
Weight (Kg)	24.0 (19; 35.9)	41 (29; 53)	<0.05	41 (20.9; 44.5)	0.145
BMI-z score	−0.06 (−0.8; 0.4)	0.49 (−0.6; 0.9)	<0.05	0.4 (−0.4; 1.2)	0230
Insulin (IU/Kg/day)	0.35 (0.1; 0.5)	0.55 (0.3; 0.7)	<0.05	-	
HbA1c (%)	7.9 (6.8; 9.1)	7.5 (6.5; 8.4)	<0.05	-	
C-peptide (ng/mL)	0.4 (0.1; 1)	0.3 (0.1; 0.9)	0.621	-	
25 OH vitamin D (ng/mL)	22.8 (17; 33.4)	27.8 (24.1; 35.8)	<0.05	-	
AA/EPA ratio	41.2 (22.2; 67.4)	36.5 (23; 68.9)	0.711	53.4 (26; 64)	0.753
Dietary assessment					
Kcal die	1724 (1222; 1896)	1784 (1511; 1952)	0.126	1565 (1435; 1769)	0.105
Sugars g/day	66.7 (46.7; 78.1)	61 (36.2; 81.6)	0.779	72.5 (54.9; 101)	0288
Sugars %	15.2 (12.6; 17.7)	13.1 (10.2; 16.8)	0.242	18.3 (15.5; 23.6)	<0.01
Lipid g/day	63 (46.9; 74.3)	67.3 (57.4; 73.9)	0.428	64.2 (47.8; 76.3)	0.415
Lipid %	33.4 (29.4; 37.9)	34.1 (30.3; 36.4)	0.646	34.2 (30.4; 38.8)	0.633
Protein g/day	67.2 (57.1; 82)	71.6 (64.9; 78.8)	0.252	53.9 (51.6; 76.2)	0.09
Protein %	16.5 (15.5; 18.7)	16.1 (14.8; 17.6)	0.476	14.6 (13.2; 17)	0.08
Fiber g/day	19.5 (11.6; 24.2)	19 (12; 22)	0.841	14.2 (10.6; 19.4)	0.248
AA g/day	0.21 (0.18; 0.23)	0.28 (0.2; 0.35)	<0.01	0.17 (0.12; 0.23)	<0.01
EPA g/day	0.15 (0.08; 0.25)	0.18 (0.1; 0.3)	0.174	0.11 (0.09; 0.2)	0.171
DHA g/day	0.32 (0.09; 0.48)	0.28 (0.05; 0.5)	0.620	0.17 (0.09; 0.34)	0.393
Chole/ergocalciferol g/day	3.63 (1.2; 6)	3.8 (1; 5.8)	0.717	2.52 (1.5; 4.3)	0.561

Group A, T1D subjects enrolled at the time of onset; Group B, T1D patients enrolled after 18 ± 5 months from onset. Data are expressed as median (IQR). Differences between groups A and B (*p* *) were compared with the nonparametric Mann–Whitney U-test. Differences between all tree groups (Groups A, B, and CS) were analyzed with Kruskal–Wallis test and post hoc analysis *p* ° group B vs. CS.

**Table 2 nutrients-14-04603-t002:** Auxologic and metabolic parameters in type 1 diabetes patients (T1D) with and without DKA at the onset (Groups A and B). NS, Non-Significant.

	T1D DKA	T1D No DKA	*p*
Number	9	31	
Age (y)	7.5 (1.8; 11.1)	9.7 (6.5; 12.5)	0.05
Weight (Kg)	34 (10.2; 44.6)	30.8 (20.2; 41.1)	NS
BMI-z score	0.8 (0.03; 1.7)	−1.0 (−1.5; −0.0)	0.05
Insulin (IU/Kg/day)	0.69 (0.57; 0.74)	0.67 (0.52; 0.78)	NS
HbA1c (%)	10.9 (10.1; 12.4)	11.8 (9.9; 13.3)	NS
HbA1c (mmol/L)	95 (87; 112)	105 (85; 124)	NS
PH (mmol/L)	7.23 (7.01; 7.26)	7.36 (7.33; 7.38)	0.001
C-peptide (ng/mL)	0.30 (0.12; 0.45)	0.44 (0.3; 0.72)	0.05
Mean Glucose (mg/dl)	439 (329; 553)	393.5 (304; 498.7)	NS
25 OH Vitamin D (ng/mL)	10.2 (5.8; 26.2)	20.5 (16.1; 28.5)	0.05

Data are expressed as medians (IQR).

**Table 3 nutrients-14-04603-t003:** Auxologic and metabolic parameters and dietary assessment in type 1 diabetes patients (T1D) enrolled at the time of onset (group A) with and without DKA.

	T1D DKA	T1D No DKA	*p*
Group A			
Number	5	15	
Age (y)	2.2 (1.1; 9.6)	8.9 (6.3; 11.2)	0.08
Weight (Kg)	12 (8.8; 39.5)	24 (20.8; 35.5)	NS
BMI-z score	0.95 (0.44; 2,9)	−0.4 (−1.1; 0.35)	0.07
AA/EPA ratio	55.8 (24; 124)	39.4 (19.4; 58.1)	NS
Kcal day	1177 (702; 1800)	1740 (1506; 1903)	NS
Sugar g/day	30.4 (14.7; 64)	71 (51.5; 79.8)	0.05
Sugar %	11 (6; 15.7)	15.9 (14.3; 17.7)	0.05
Lipid g/day	48.9 (24.5; 65.3) T1D	64 (50.6; 78.5)	NS
Lipid %	32.8 (28; 41.2)	34.3 (30.3; 38.6)	NS
Protein g/day	57.4 (29.8; 68.4)	70.5 (61.4; 87.2)	0.05
Protein %	15.5 (15.2; 18.6)	16.5 (15.7; 18.9)	NS
Fiber g/day	12.5 (5.5; 19.5)	20.5 (14.8; 25.8)	0.05
AA g/day	0.2 (0.12; 0.22)	0.21 (0.18; 0.24)	NS
EPA g/day	0.07 (0.04; 0.1)	0.2 (0.13; 0.3)	0.01
DHA g/day	0.09 (0.05; 0.14)	0.38 (0.29; 0.51)	0.01
Vitamin D µg/day	1 (0.6; 1.4)	5.7 (3.1; 7.2)	0.01

Data are expressed as medians (IQR).

## Data Availability

Not applicable.

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
