# Peer review of "Vitamin D Repletion and AA/EPA Intake in Children with Type 1 Diabetes: Influences on Metabolic Status"

_nutrients, 2022, doi:10.3390/nu14214603_

Round 1

Reviewer 1 Report

This is a study aimed to show a relationship between metabolic control, Vitamin D status (25OHD), and arachidonic acid (AA)/eicosapentaenoic acid (EPA) ratio in children with Type 1 Diabetes (T1D). The authors had a secondary aim, which was to evaluate dietary intake and the presence of ketoacidosis (DKA) at the onset of T1D.

Major comments:

In the section of Materials and Methods, the author mentions that subjects with celiac disease or thyroiditis were not excluded. It is important to describe how many subjects had celiac disease and how many thyroiditis because the first disease is of clear autoimmune origin, and the diet received by patients with celiac disease is modified by avoiding certain food groups; this could be biasing the results observed both in dietary intake and serum levels of vitamin D. In the case of patients with thyroiditis, the authors do not mention whether it is secondary or primary (of autoimmune origin), this pathology, regardless of the origin, has a strong impact on the modification of metabolites of vitamin D, such as calcidiol and calcitriol. Describe in results how many patients presented these pathologies and exclude them from the analyses, repeat them to describe if the significance of the reported results is maintained only in patients with T1D or could be influenced by these conditions.

In the Results section, the authors mention that differences in 25OHD levels were detected when comparing the study groups despite similar dietary intake of cholecalciferol/ergocalciferol. Describe in the Discussion section why the impact of diet on vitamin D levels is insignificant.

The authors must describe in the discussion section the limitations of the study supporting the following points: 1) at the methodological level, 2) study design, 3) establishment of causal relationships or associations, as well as 4) the perspectives of the study. 

Author Response

This is a study aimed to show a relationship between metabolic control, Vitamin D status (25OHD), and arachidonic acid (AA)/eicosapentaenoic acid (EPA) ratio in children with Type 1 Diabetes (T1D). The authors had a secondary aim, which was to evaluate dietary intake and the presence of ketoacidosis (DKA) at the onset of T1D. 

Major comments:

In the section of Materials and Methods, the author mentions that subjects with celiac disease or thyroiditis were not excluded. It is important to describe how many subjects had celiac disease and how many thyroiditis because the first disease is of clear autoimmune origin, and the diet received by patients with celiac disease is modified by avoiding certain food groups; this could be biasing the results observed both in dietary intake and serum levels of vitamin D. In the case of patients with thyroiditis, the authors do not mention whether it is secondary or primary (of autoimmune origin), this pathology, regardless of the origin, has a strong impact on the modification of metabolites of vitamin D, such as calcidiol and calcitriol.

Describe in results how many patients presented these pathologies and exclude them from the analyses, repeat them to describe if the significance of the reported results is maintained only in patients with T1D or could be influenced by these conditions.

Thanks for your comment. We have analyzed without these 5 patients and found no significant differences in any of the variables, including nutrient intake at baseline, so we decided to include them again without modifying Table 1.

However, as suggested, we have clarified the frequencies of patients with celiac disease or hypothyroidism in the results, and we pointed out in the text that their exclusion did not modify the significance.

For your convenience, you can find below the data table without these 5 patients.

T1D

Group A

T1D

Group B

CS

Number

18

17

20

Age (ys)

8.1 (5.9; 11)a

9.95 (7.9; 13)a

   11.5 (9.3; 13.6)

Weight (Kg)

24.0 (19.3; 37.4)a

38 (28.5; 48.5)a

41 (20.9; 44.5)

BMI-z score

-0.06 (-0.5; 0.5)

0.31 (-0.9; 0.9)

0.4 (-0.4; 1.2)

Insulin (IU/Kg/day)

0.35 (0.1; 0.4) a

0.52 (0.3; 0.8)a

-

HbA1c (%)

7.9 (6.7; 8.9)

7.7 (6.5; 8.4)

-

C-peptide (ng/mL)

0.5 (0.1; 1.1)

0.36 (0.1; 0.9)

-

25 OH vitamin D (ng/mL)

23.4 (17.4; 32.3)a

27.1 (23.3; 33.9)a

-

AA/EPA ratio

39.8 (19.4; 69.6)

34.8 (22.4; 63.1)

53.4 (26; 64)

Dietary assessment

Kcal die

1724 (1265; 1886)

1809 (1605; 1957)

1565 (1435; 1769)

Sugars g/die

66.7 (47; 77.9)

63 (39.8; 86.6)

72.5 (54.9; 101)

Sugars %

15.2 (13.3; 17.7)

14 (10.1; 17.7)b

18.3 (15.5; 23.6)b

Lipid g/die

64 (46.1; 75.4)

70 (57.5; 75.6)

64.2 (47.8; 76.3)

Lipid %

33.4 (30.3; 38.6)

34.0 (30.2; 36.3)

34.2 (30.4; 38.8)

Protein g/die

67.2 (55.3; 79.8)

72.4 (69; 83)

53.9 (51.6.; 76.2)

Protein %

16.2 (15.5; 18.0)

16.1 (15.1; 18.2)

14.6 (13.2; 17)

Fiber g/die

19.5 (12.2; 24.0)

20 (14.6; 24)

14.2 (10.6; 19.4)

AA g/die

0.21 (0.17; 0.23)a

0.28 (0.2; 0.35)a,b

0.17 (0.12; 0.23)b

EPA g/die

0.15 (0.08; 0.27)

0.21 (0.1; 0.3)

0.11 (0.09; 0.2)

DHA g/die

0.32 (0.11; 0.5)

0.3 (0.1; 0.5)

0.17 (0.09; 0.34)

Chole/ergocalciferol g/die

3.63 (1.4; 5.9)

4 (1.5; 5.9)

2.52 (1.5; 4.3)

Table 1. Auxologic, metabolic parameters and dietary assessment in type 1 diabetes patients (T1D) and control subjects (CS) at the enrolment (T0). Group A: T1D subjects enrolled at the time of onset; Group B: T1D patients enrolled after 18±5 months from onset.

                     Data are expressed as median (IQR).

Differences between groups A and B  were compared with the nonparametric Mann-Whitney U-test; a =p<0.05 T1D group A vs B

Differences between all tree groups (group A, B, CS) were analyzed with  Kruskall-Wallis test and post hoc analysis;  b=p<0.01 T1D group  B vs CS

In the Results section, the authors mention that differences in 25OHD levels were detected when comparing the study groups despite similar dietary intake of cholecalciferol/ergocalciferol. Describe in the Discussion section why the impact of diet on vitamin D levels is insignificant.

Thanks for your suggestion. We have added this point in the discussion section.

The authors must describe in the discussion section the limitations of the study supporting the following points: 1) at the methodological level, 2) study design, 3) establishment of causal relationships or associations, as well as 4) the perspectives of the study. 

Thanks for your comments. We add these points in discussion as suggested.

Reviewer 2 Report

The study’s results show interesting associations of vitamin D status and Omega 3 fatty acids intake among children with T1D with severity of DKA, C-peptide, daily insulin requirements and some dietary intakes. This knowledge could be used when forming individual treatment goals and recommendations in children with T1D.

The study design, results and discussion are clear.

I have only some concerns:

In the Materials and Methods: The Authors enrolled only children with quite short duration of diabetes, <5years. I am wondering what was the rationale for this cut-off point? Moreover, what was the rationale for further subdivisions into the group <2months of T1D duration and >18months of T1D duration?

In Table 3 : The term “Gruppo A” should be changed into “Group A”

In the text: the daily insulin requirements units are “IU/Kg/die”, perhaps should be changed into “IU/Kg/day” to unify the language.

Author Response

The study’s results show interesting associations of vitamin D status and Omega 3 fatty acids intake among children with T1D with severity of DKA, C-peptide, daily insulin requirements and some dietary intakes. This knowledge could be used when forming individual treatment goals and recommendations in children with T1D.

The study design, results and discussion are clear.

I have only some concerns:

In the Materials and Methods: The Authors enrolled only children with quite short duration of diabetes, <5years. I am wondering what was the rationale for this cut-off point? Moreover, what was the rationale for further subdivisions into the group <2months of T1D duration and >18months of T1D duration?

Thanks for your comment. Now, we have better explained our inclusion criteria. They are linked to the fact that these date are the proof of concept study that was used to design the study published two years ago (Cadario F. Nutrients 2019, 11(9), 2158; https://doi.org/10.3390/nu11092158).

Moreover, we add these points in discussion section in the limits of the study.

In Table 3 : The term “Gruppo A” should be changed into “Group A”.

Thanks. Done

In the text: the daily insulin requirements units are “IU/Kg/die”, perhaps should be changed into “IU/Kg/day” to unify the language.

Thanks. Done

Reviewer 3 Report

The authors aimed to document: A) The causal relationship (if any) between vitamin D status [25OHD, (ng, mL-1): Sufficient (≥30), insufficient (20-29), deficient (20)], blood arachidonic/eicosapentanoic (AA/EPA) acid ratio [dry blood spots; cutoff 22.5 (first quartile)], diet quality intake (including Vit D & AA/EPA) with certain physiological/glycemic homeostatic parameters in a pediatric cohort (n= 40, ~9.7y) with onset (group A, n=20; ~8.1y) and late (18m, group B; n=20; ~10.5y) diagnosis of type 1-diabetes (T1D) and compared to healthy control subjects (Group C; n= 20; 11.5y; just sociodemographic and diet intake data), and B) To evaluate dietary intake and the presence of ketoacidosis (DKA) at the onset of T1D. Group-specific differences were documented for BMI (z-score; group A<B,C) sugar (C>A>B) and AA (B>A>C) intake. Insulin and 25OHD levels were influenced by AA/EPA status. DKA children (n= 9) showed a higher BMI and lower insulin, pH, c-peptide and 25OHD serum levels, lower sugar, AA, EPA, and vit D intake as compared to non-DKA children (n=31). Lastly, Vit D status showed a cubic relationship (trend, r~1.0) with pH (+X3) and C-peptide (-X3). The experimental design and execution and the evidence-based discussion are the strongest points of this study. However, it is recommended to consider the following to improve the scientific soundness and uniqueness (differentiation) of the study:  

General

·         The manuscript will improve even more if the English grammar and style are reviewed once again by a native English-spoken person or by a formal translation agency.

·         The meaning of all abbreviations should be clarified the first time they are mentioned (e.g. Th17 and ICOS+, line 70) and reduce those unneeded [e.g. M 19/F 21 = male (19), female(21)] or wrong (e.g. Ω-3 are not all PUFAs; line 54, 78) ones as much as possible.

Title. Quite long. Suggestion= Vitamin D and AA/EPA status in children with type 1 diabetes: From disease onset to metabolic acidosis.

Abstract. OK. Just be aware that linear “r” values (~0.37) could improve if exploring the goodness-of-fit method to your data (preliminary, I detected a X3-polynomial relationship).

Introduction. This section should highlight the scientific contribution of the study to a topic that has apparently been widely visited.    

Methods. OK

Results & Discussion. OK.

Tables. Please include statistically significant differences between groups in table 1

Figures. In general, all figures should be of a higher resolution (>300 dpi).

References. Check once again for any unformatted or not properly cited reference and if possible reduce to say 40.

Author Response

The authors aimed to document: A) The causal relationship (if any) between vitamin D status [25OHD, (ng, mL-1): Sufficient (≥30), insufficient (20-29), deficient (20)], blood arachidonic/eicosapentanoic (AA/EPA) acid ratio [dry blood spots; cutoff 22.5 (first quartile)], diet quality intake (including Vit D & AA/EPA) with certain physiological/glycemic homeostatic parameters in a pediatric cohort (n= 40, ~9.7y) with onset (group A, n=20; ~8.1y) and late (18m, group B; n=20; ~10.5y) diagnosis of type 1-diabetes (T1D) and compared to healthy control subjects (Group C; n= 20; 11.5y; just sociodemographic and diet intake data), and B) To evaluate dietary intake and the presence of ketoacidosis (DKA) at the onset of T1D. Group-specific differences were documented for BMI (z-score; group A<B,C) sugar (C>A>B) and AA (B>A>C) intake. Insulin and 25OHD levels were influenced by AA/EPA status. DKA children (n= 9) showed a higher BMI and lower insulin, pH, c-peptide and 25OHD serum levels, lower sugar, AA, EPA, and vit D intake as compared to non-DKA children (n=31). Lastly, Vit D status showed a cubic relationship (trend, r~1.0) with pH (+X3) and C-peptide (-X3). The experimental design and execution and the evidence-based discussion are the strongest points of this study. However, it is recommended to consider the following to improve the scientific soundness and uniqueness (differentiation) of the study:  

General

  • The manuscript will improve even more if the English grammar and style are reviewed once again by a native English-spoken person or by a formal translation agency.

Thanks for your suggestion. We have carefully revised English with the help of a native English-speaking colleague.

  • The meaning of all abbreviations should be clarified the first time they are mentioned (e.g. Th17 and ICOS+, line 70) and reduce those unneeded [e.g. M 19/F 21 = male (19), female(21)] or wrong (e.g. Ω-3 are not all PUFAs; line 54, 78) ones as much as possible.

Thanks. Done.

Title. Quite long. Suggestion= Vitamin D and AA/EPA status in children with type 1 diabetes: From disease onset to metabolic acidosis.

Thanks for your suggestion. We modify the title in “Vitamin D repletion, and AA/EPA Omega 3 fatty acids intake in children with Type 1 Diabetes: influences on metabolic status”.

Abstract. OK. Just be aware that linear “r” values (~0.37) could improve if exploring the goodness-of-fit method to your data (preliminary, I detected a X3-polynomial relationship).

Thank you for the suggestion. The first one was cubic, the second linear. We have modified the text.

Introduction. This section should highlight the scientific contribution of the study to a topic that has apparently been widely visited. 

Thank you for your comment. We have inserted some sentences in the introduction and discussion to explain more the contribution of our study in the field.

Methods. OK Thanks.

Results & Discussion. OK. Thanks.

Tables. Please include statistically significant differences between groups in table 1

Thanks. Done

Figures. In general, all figures should be of a higher resolution (>300 dpi).

Done.

References. Check once again for any unformatted or not properly cited reference and if possible reduce to say 40.

Thanks for your comment. We have reduced the bibliography by removing some references. However, we have added others to respond to some reviewer’s comments.

Round 2

Reviewer 1 Report

The authors have made the suggested modifications and clarified the doubtful points that were questioned in the first round of comments.

It is suggested to check the grammar with a native English speaker.